# Differential Responses of Bacterial and Fungal Communities to Siderophore Supplementation in Soil Affected by Tobacco Bacterial Wilt (*Ralstonia solanacearum*)

**DOI:** 10.3390/microorganisms11061535

**Published:** 2023-06-09

**Authors:** Yunxin Shen, Jiangyuan Zhao, Xuefeng Zou, Zhufeng Shi, Yongqin Liao, Yonghong He, Hang Wang, Qibin Chen, Peiweng Yang, Minggang Li

**Affiliations:** 1College of Plant Protection, Yunnan Agricultural University, Kunming 655508, China; yxshen1996@126.com (Y.S.); qq10540215100@sina.com (X.Z.); lyq997480@outlook.com (Y.L.); heyonghong700327@163.com (Y.H.); tclass99@163.com (Q.C.); 2Institute of Agricultural Environment and Resources, Yunnan Academy of Agricultural Sciences, Kunming 650204, China; shizhfe@163.com; 3Yunnan Institute of Microbiology, Yunnan University, Kunming 650106, China; jyzhao@ynu.edu.cn; 4National Plateau Wetlands Research Center, Wetlands College, Southwest Forestry University, Kunming 650233, China; hwang17@163.com

**Keywords:** siderophore, tobacco bacterial wilt, bacterial community, fungi, agricultural soil

## Abstract

Siderophores secreted by microorganisms can promote ecological efficiency and could be used to regulate the unbalanced microbial community structure. The influence of the siderophore activity of Trichoderma yunnanense strain 2-14F2 and Beauveria pseudobassiana strain (2-8F2) on the physiological/biochemical functions and community structure of soil microbes affected by tobacco bacterial wilt (TBW) was studied. DNS Colorimetry and Biolog-eco plates were used to quantify the impacts of strain siderophores on soil enzyme activities and microbial metabolism. Based on Illumina MiSeq high-throughput sequencing, the soil 16S rDNA and ITS sequences were amplified to dissect the response characteristics of alpha/beta diversity and the structure/composition of a soil microbial community toward siderophores. The KEGG database was used to perform the PICRUSt functional prediction of the microbial community. We found that siderophores of 2-14F2 and 2-8F2, at certain concentrations, significantly increased the activities of sucrase (S-SC) and urease (S-UE) in the TBW soil and enhanced the average well color development (AWCD, carbon source utilization capacity) of the microbial community. The metabolic capacity of the diseased soil to amino acids, carbohydrates, polymers, aromatics, and carboxylic acids also increased significantly. The response of the bacterial community to siderophore active metabolites was more significant in alpha diversity, while the beta diversity of the fungal community responded more positively to siderophores. The relative abundance of *Actinobacteria*, *Chloroflexi*, and *Acidobacteria* increased and was accompanied by reductions in *Proteobacteria* and *Firmicutes*. LEfSe analysis showed that *Pseudonocardiaceae*, *Gemmatimonas*, *Castellaniella*, *Chloridiumand* and *Acrophialophora* altered the most under different concentrations of siderophore active metabolites. The PICRUSt functional prediction results showed that siderophore increased the abundance of the redox-related enzymes of the microbial community in TBW soil. The BugBase phenotypic prediction results showed that the siderophore activity could decrease the abundance of pathogenic bacteria. The study concludes that siderophore activity could decrease the abundance of pathogenic bacteria and regulate the composition of the microbial community in TBW soil. The activities of sucrase (S-SC) and urease (S-UE) in TBW soil were significantly increased. Overall, the siderophore regulation of community structures is a sustainable management strategy for soil ecosystems.

## 1. Introduction

The rhizosphere is the key microdomain of plant–soil–microbe interactions. There are about 10^9^ microorganisms colonized in each gram of rhizosphere soil [1]. The occurrence of soil-borne diseases mainly depends on the interaction of pathogens and other factors in the key microdomain. Tobacco bacterial wilt (TBW) is one of the most important soil-borne diseases, which can reduce tobacco production by more than 50% or even eliminate the harvest, bringing great obstacles to tobacco production [2]. The occurrence of TBW depends on the influence of *Ralstonia solanacearum* colonizing in tobacco rhizosphere soil in key microdomains [3]. When the abundance of the pathogen population reaches a threshold, it can invade from the tobacco rhizosphere and cause plant disease [4]. At present, chemical control measures are mainly used for TBW. Although it can be timely and effective, the control outcome is limited, and the unbalanced soil microbial community structure cannot be adjusted [5]. Therefore, from the perspective of ecological balance, it is of great significance to use biocontrol methods against TBW. Presently, the use of SPMs (siderophore-producing microorganisms) to regulate the unbalanced soil microbial community structure has become an effective biocontrol measure [6].

In the natural environment, soil and crust are rich in iron, an essential element for plants and microorganisms [7]. Iron plays a vital role in the process of information transmission and physiological metabolism in cells [8]. However, naturally occurring iron is mostly in the form of relatively stable Fe^3+^ due to oxidation, and the content of soluble Fe^2+^ is very low in the soil. In agricultural production, plant iron deficiency is very common, which limits the growth and development of plants, declines their systemic resistance, and hinders increases in crop yields [9]. Therefore, to adapt to an environment with extremely low soluble iron content, soil microorganisms secrete small molecule compounds, siderophores, with a specific chelating force for Fe^3+^ in the environment via secondary metabolism [10]. The complex of siderophore and iron ions can also be absorbed by microbes or plants, which can provide enough iron for organisms. According to the different structural types of siderophore compounds secreted by the strains, they are divided into hydroxamic salt type, catechol salt type, carboxylate type, etc., and the hydroxamic salt type and carboxylate type can be secreted by the fungi. SPMs can be used to adjust the imbalance of the soil microbial community structure, which is based on the fact that siderophore-producing strains (SPSs) have a certain competitive advantage for iron in the soil microbial community [11]. So far, there are some reports on the antibacterial effect of siderophore and SPSs in biocontrol [12]. Previous studies have found that bacteria with siderophore production have more significant inhibition on pathogenic microorganisms than bacteria without siderophore production, which can reduce the number of pathogenic bacteria cells and inhibit the formation of pathogenic microbial biofilms [13,14,15]. In addition, based on the iron chelation principle of microbial siderophores, the microbial community model under the intervention of siderophores was established, and the SPMs are thought to have absolute advantages in the competition for iron ions in the soil microbial community, by which the community structure of plant pathogens is governed [16].

To date, more than 500 different types of siderophores have been found, of which 270 are characterized in their structures [17]. The use of siderophores secreted by microbes is the main measure used to develop new iron supplements and iron additives and improve the utilization rate of iron. However, the number of siderophores for which we have a clear understanding is still very limited (less than ten types), and there is a lack of siderophores that better reflect the strain specificity [18]. Fortunately, siderophore active substances in the microbial communities of primary forests or agricultural soils or oceans can specifically bind iron ions in the environment. We hypothesize that, based on the biology and environment of the original forest in Ailao Mountain, the exploration and utilization of the microbial resources of siderophores can provide the essential species of and genetic, physiological and biochemical information about siderophores. In this study, two strains of secretory siderophore fungi were isolated from the soil of a primitive forest in Ailao Mountain, and the effects of their siderophores on the physiological/biochemical functions and community structure of TBW soil microorganisms were investigated in detail. The aim is to use microbial secreted siderophores to regulate a TBW soil ecosystem and provide a healthy and sustainable strategy for TBW prevention and control.

The Ailao Mountain primeval forest of Yunnan Province, a National Nature Reserve, is located in the southeast of the Qinghai Tibet Plateau (QTP) and the famous longitudinal valley of the Hengduan Mountains. The East Asian monsoon climate, QTP monsoon climate, and tropical monsoon climate of South and Southeast Asia gather here. Due to the coupling of the special terrain, climate, and hydrological characteristics, the soil is rich in organic matter, and there are abundant species and germplasm resources with great economic value [19]. According to the results of the isolation and screening of soil microbes, and antagonism experiments on them, the soil environment of the primeval forest in this region is rich in SPMs, which makes it worthy of further exploration. As a precious natural gift, the siderophore microbe is closely related to the sustainable development of agriculture [20]. It plays an extremely important role in the improvement and maintenance of soil fertility, the transformation of nutrient elements, the prevention and control of crop diseases, environmental remediation, and the balance of the ecosystem [21]. It is an important foundation for the research, development, and utilization of innovative drugs and environmental protection preparations, including, e.g., microbial fertilizer, microbial pesticide, microbial food, microbial feed, environmental hormone, environmental engineering microbe, etc. [22].

## 2. Material and Methods

### 2.1. Sample Sources

The secretory siderophore fungi were isolated from the forest soil (100°49′–101°53′ E, 23° 95′–24°17′ N, altitude 2480–2486 m) in Ailaoshan National Nature Reserve of Yunnan Province. The area has a subtropical monsoon climate with an average annual temperature of 18.9–22.6 °C and rainfall of 2200 mm. In 0–20 cm soil, the organic matter content was 47.58–754.78 g·kg^−1^, the water content was 9.60–42.70%, and the iron mass content was 0.54–1.18%, with a pH of 4.54–6.71.

The rhizosphere soil affected by TBW was collected from Datianba Town, Changning County, Baoshan City, Yunnan Province. The tobacco planting base (98°32′–99°63′ E, 24°92′–23°43′ N, altitude 2000 m) has a subtropical monsoon climate, with an average annual temperature of 15–17 °C and an average annual rainfall of 1290 mm. In 0–20 cm soil, the organic matter content was 28.54 g·kg^−1^, and the available nitrogen, available phosphorus, and available potassium were 134 mg·kg^−1^, 27 mg·kg^−1^, and 167 mg·kg^−1^, respectively, with a pH of 4.71, according to the standard test methods [23].

Fifteen soil samples of tobacco bacterial wilt (TBW) and thirty soil samples of the forest were selected. We mixed the soil samples and divided the mixture into 50 mL centrifuge tubes according to the following protocol: add 2 g of a soil sample to 198 mL of sterile water (500 mL conical flask with glass beads); shake at 25 °C (180 rpm) for 4–6 h; let stand for 2–3 min; dilute supernatant gradient to 10^−2^, 10^−3^, and 10^−4^; apply 0.2 mL to plate; repeat twice for each sample. After the colony grew on the plate, a single colony with a distinct morphology was selected (repeatedly) for the culture medium. The impure strains also needed to be purified on the plate, and the purified strains were inoculated on the culture medium, and the fungi were cultured at 28 °C for 5–10 days. The experiments were repeated 3 times.

### 2.2. Fungal Strains with a High Yield of Siderophore

*Trichoderma yunnanense* (strain No. 2-14F2) and *Beauveria pseudobassiana* (strain No. 2-8F2) were isolated from the soil samples collected in Ailaoshan National Nature Reserve and were deposited in the China General Microbiological Culture Collection Center (CGMCC), with the preservation numbers of CGMCC No. 17926 and CGMCC No. 21047.

### 2.3. Preparation of the Test Solution and Culture Medium

A total of 0.0605 g of chromeazurol S (CAS) was dissolved in 50 mL of deionized water, 10 mL FeCl_3_ solution was added, and they were stirred and mixed. This was labeled as “solution A”. Then, 0.0729 g of cetyltrimethylammonium bromide was dissolved in 40 mL deionized water, and this was labeled as “solution B”. Then, solution A was slowly poured into solution B, and the mixture was stirred evenly.

Other media for strain culture included the following: PDA medium: potato 200 mg·mL^−1^; glucose 20 mg·mL^−1^; agar 15–20 g; deionized water 1000 mL; pH 7.0 ± 0.1. Iron-free Czarist’s liquid medium: glucose 30.0 mg·mL^−1^; sodium nitrate 2.0 mg·mL^−1^; potassium phosphate trihydrate 1.0 mg·mL^−1^; potassium chloride 0.5 mg·mL^−1^; magnesium sulfate heptahydrate 0.5 mg·mL^−1^; 8-hydroxyquinoline 0.75 mg·mL^−1^; deionized water 1000 mL; pH 7.0 ± 0.1. NA medium: peptone 10.0 mg·mL^−1^; beef powder 3.0 mg·mL^−1^; sodium chloride 5.0 mg·mL^−1^; agar 15.0 mg·mL^−1^; pH 7.3 ± 0.1. Double-layer color medium: bottom layer 5 mL.

### 2.4. Morphological Observation and Molecular Characterization

For morphological observation, the strain was cultivated for 7 days on Petri dishes, and the fungal mycelia (3 mm × 3 mm) was cut off with a double-sided blade, fixed with 1% glutaraldehyde for 30 min, cleaned with distilled water twice (20 min/time), and dehydrated with different serial concentrations of ethanol (50%, 70%, 80%, 90%, and 100%) and butanol (75% and 100%) for 15 min. Then, the fungal mycelia were solidified in the refrigerator freezer and dried by vacuum sublimation, and gold was sprayed on fungal mycelia by IB-3 ion coating instrument. The gold-coated sample was then observed by FEI Quanta 200 scanning electron microscope (SEM).

DNA was extracted from 2-8F2 and 2-14F2 mycelia preparation. Fungi ITS genes were amplified with the universal ITS1 (5′-TCCGTAGGTGAACCTGCGG-3′) and ITS4 (5′-TCCTCCGCTTATTGATATGC-3′). The PCR system was as follows: 10 × Buffer 2.5 μL, dNTP (2.5 mmol·L^−1^) 4 μL, 1.0 μL MgCl_2_ (25 mmol·L^−1^), 1.0 μL template DNA, 1.0 μL ITS1 primer (10 μmol·L^−1^), 1.0 μL ITS4 primer (10 μmol·L^−1^), 1.0 μL Taq polymerase (2.5 U·μL^−1^), and ddH2O 17 μL. PCR reaction conditions included pre-denaturation at 94 °C for 5 min; 50 cycles of denaturation at 94 °C for 1 min, annealing at 56 °C for 30 s, and extending for 1 min at 72 °C; extending for 10 min at 72 °C and preserving at 4 °C. The base deletion sites were excluded in phylogenetic analysis, and the phylogenetic tree of tested strains and reference strains were constructed by neighbor-joining, maximum-parsimony, and maximum-likelihood analysis approaches using MEGA 7. The bootstrap value was set to 1000.

### 2.5. Determination of Secretory Activity and Type Identification of Siderophore

The strains were inoculated into Tiecha-free liquid medium and cultured in a constant temperature shaker (HS-200B, Huanan Xinhai, Shenzhen, China) at 28 °C with 150 r/min for 48 h. After the culture, 2 to 5 mL of culture medium was absorbed and filtered by 0.22 μm sterile filter membrane, and an equal volume of CAS detection solution was added. After standing for 1 h, full-wavelength enzyme labeler (model: Multiska GO) was used to measure OD 630 (denoted as “AS”), and OD630 of uninoculated liquid medium (denoted as “Ar”) was measured by the same method. The concentration of siderophore was expressed by siderophore unit (SU): SU = [(AR − AS)/Ar] × 100%. The determination was repeated 3 times, and the mean value was taken for comparative analysis.

The structure type of siderophore in the above-mentioned culture liquid was determined by the method of the FeCl_3_ test [24]. The spectrophotometric data were recorded on a Multiskan GO full-wavelength microplate reader (Thermo Scientific, Waltham, MA, USA). Ferric hydroxamate absorbs light at 420–450 nm, while a 495 nm peak of ferrated siderophore indicates the presence of a catecholate siderophore.

### 2.6. Separation of Siderophore

We put 25 g of steamed rice into a 500 mL tissue culture flask, sterilized it under high pressure at 121 °C for 30 min, added 1 mL of 2-8F2 or 2-14F2 bacterial suspension, stirred thoroughly, and fermented for 30 days in a 37 ± 1 °C constant temperature incubator. After completing the fermentation, we soaked it thoroughly in 95% methanol for 24 h and filtered it with 5 mm aperture filter paper. Finally, we used a rotary evaporator (R-210 BUCHI, Flawil, Switzerland) at 45 °C with 700 r/min to completely evaporate the organic solvent in the filtrate.

We used column chromatography to separate the active substances. Firstly, a preliminary segmented separation was carried out using normal phase silica gel G (200–300 mesh, Qingdao Marine Chemical Industrial Products Factory, Qingdao, China), and a mobile phase gradient constant velocity elution was used according to the method of Pan et al. [25]. Sephadex-LH20 gel (GE, Boston, MA, USA) was used to continue purification of the residue separated by normal phase silica gel. The mobile phase is methanol, and the flow rate is 10 mL/30 min. A total of 10 mL of each fraction of the residue was collected by an automatic collector. CAS liquid and 2% FeCl_3_ were used to detect the color development of each fraction. The fraction that reacts with CAS liquid and turns red and the fraction that reacts with 2% FeCl_3_ and turns brown or tan are the active fraction. According to the color development, the active fraction is combined and concentrated for standby.

The obtained iron carrier active fraction was evaporated to dryness under reduced pressure and stored in a refrigerator at 4 °C. When used, we dissolved it in deionized water and prepared solutions of 0.05 mg·mL^−1^, 0.15 mg·mL^−1^, and 0.45 mg·mL^−1^ for future use.

### 2.7. Treatment of TBW Soil

The fresh soil samples in the TBW rhizosphere were selected. A total of 30 g of soil samples were weighed and poured into 50 mL centrifugal pipe, and 5 mL of distilled water was added. After 5 min, the siderophore containing active parts of 2-8F2 and 2-14F2 strains were added. The CK group (5 mL distilled water) and treatment group (5 mL siderophore compound) were set up. The two strains with the concentration of 0.05 mg·mL^−1^ were labelled as 2-8F2 A and 2-14F2 A. The strains with the concentration of 0.15 mg·mL^−1^ were labelled as 2-8F2 B and 2-14F2 B, and the strains with the concentration of 0.45 mg·mL^−1^ were 2-8F2 C and 2-14F2 C, respectively. Each treatment was dark-cultured in 37 ± 1 °C incubator for 30 days.

The classification standard of TBW [26] was adopted in the investigation of diseased soil. Grade 0 represented no symptoms in leaves; Grade 1 represented that less than 1/4 of leaves showed wilting symptoms; Grade 2 represented that 1/4–1/2 leaves showed wilting symptoms; Grade 3 represented that more than 1/2 of leaves showed wilting symptoms; Grade 4 represented the wilting death of the whole plant. Disease index = ∑(Number of diseased plants at all levels × Corresponding level)/(Total number of investigated plants × highest level) × 100.

### 2.8. DNA Extraction, Illumina MiSeq High-Throughput Sequencing, and Bioinformatics/Statistical Analysis

The genomic DNA of soil microorganisms was extracted by the OMEGA total DNA extraction kit according to the instruction manual. The quality of extracted DNA was detected by 0.8% agarose gel electrophoresis, and DNA was quantified by ultraviolet spectrophotometer. The qualified DNA was sent to MajorBio Biotechnology Co., Ltd. (Shanghai, China). for high-throughput amplicon sequencing. 16S rRNA gene of bacteria and ITS sequence of fungi were amplified by PCR. The V3–V4 region of bacteria (338F, 5′-ACTCCTACGGGAGGCAGCAG-3′; 860R, 5′-GGACTACHVGGGTWCTAAT-3′) and ITS1 region of fungi (ITS1F, 5′-CTTGGTCATTTAGAGGAAGTAA-3′; 2043R, 5′-GCTGCGTTCTTCATCGATGC-3′) were PCR amplified. The amplified products were used as templates for the preparation of the Illumina MiSeq sequencing library, which was subject to high-throughput sequencing. The representative sequences of each OTU (operational taxonomic unit) were obtained by BLAST search against NCBI GenBank, and then multiple sequence alignment was performed by DNAMAN based on the translated amino acid sequences. The neighbor-joining phylogenetic tree was constructed in MEGA 7 for sequences for which abundance was more than 1.0% in at least one sample, and 1000 bootstrap replicates were set to obtain the support value of each branch.

The problematic sequences were identified from raw sequencing data, which were checked, and the chimeric sequences were removed to obtain the effective sequences of samples. By taking the most abundant sequence in each OTU as the representative sequence of the OTU, OTU was merged and divided based on 97% sequence similarity, the OTU abundance matrix in each sample was constructed, and the total number of OTUs in each sample was calculated. The microbial community richness was evaluated by the Chao1 index, and the community diversity was evaluated by the Shannon index. The template sequence of the corresponding database was compared with the representative sequence of each OTU, and the corresponding taxonomy information of each OTU was obtained. The community composition of each sample at each taxonomic level was counted. In the cloud platform of MajorBio Biotechnology Co., Ltd. (cloud.majorbio.com, accessed on 12 August 2021), the species annotation and assessment, e.g., alpha diversity, and species composition analysis were carried out. The non-parametric factorial Kruskal–Wallis (KW) sum-rank test was used to detect the significant difference of relative abundance. Then, LEfSe was used in the linear discriminant analysis (LDA threshold ≥ 2) to estimate the influence of abundance of each component (species) on the different effects.

In PICRUSt, which stores the COG information and KO (KEGG ortholog) information corresponding to Greengene ID, the OTU abundance tables were standardized; i.e., the influence of the copy number of 16S rRNA gene in the species genome was eradicated. Then, the COG family information and KO information were obtained via the Greengene ID corresponding to each OTU, and the COG abundance and KO abundance were calculated. Based on the COG database, the descriptive information of each COG and its functional information can be parsed from the eggNOG database to obtain the functional abundance spectrum. In the KEGG database (http://www.genome.jp/kegg/, accessed on 22 August 2021), the enzyme function of the soil microbial community was predicted and analyzed. In BugBase (https://bugbase.cs.umn.edu/index.html, accessed on 26 August 2021), OTU was normalized by the predicted 16S rDNA copy number, and then the microbial phenotype was predicted using the provided pre-calculated file. The predicted phenotypes include Gram-positive, Gram-negative, biofilm forming, pathogenic, mobile-element-containing, oxygen-utilizing (aerobic, anaerobic, and facultatively anaerobic), oxidative stress tolerant, etc.

### 2.9. Soil Sucrase/Urease Activities and Soil Biolog

The fresh soil samples of each treatment were freeze-dried and rough-ground and then screened by a 40 mesh screening. The second grounding was performed, followed by a 60 mesh screening. The 0.1 g and 0.2 g soil samples were weighed, and soil sucrase (S-SC) and urease (S-UE) were determined by DNS colorimetry. The kit of Suzhou Grace Biotechnology Co., Ltd (Suzhou, China). was used according to the manufacturer’s instructions. After reagents were added sequentially, the mixture was incubated in a 37 °C water bath. After incubation, the absorbance values of S-SC and S-UE were read at 540 nm and 578 nm, respectively. S-SC activity (mg/d/g) = 2.2 × (ΔA + 0.0445)/W × D; S-UE activity (mg/d/g) = 27.6 × (ΔA + 0.0051)/W. ΔA = A_measuring tube_ − A_control tube_; W was the actual weight of soil sample; and D was the dilution multiple. Average well color development (AWCD) index = ∑(31, i = 1)(*C_i_* − *R*)/31, *C_i_* was the optical density difference between the 590 nm and 750 nm bands of the I carbon source hole. *R* was the optical density value of the control hole, and 31 was the number of carbon sources tested by ECO plate. 

The Biolog GN (Grace, Suzhou, China) plate was used to determine the physiological distribution of soil microbes at the community level. Soil microbial community diversity was measured by Biolog-ECO method: Weigh 10 g of fresh soil in a 250 mL triangular bottle and add 90 mL 0.85% NaCl solution. Shaken at 25 °C for 30 min at 200 r·min^−1^. Add the supernatant to 9 mL NaCl solution and dilute it in equal proportions to 1꞉1000. Add 150 μL diluent to each well of ECO microplate. The inoculated Biolog-ECO plate was cultured in a dark environment at 25 °C for 8 days. Its absorbance was measured on a Biolog microplate readout every 24 h.

## 3. Results

### 3.1. Field Investigation of TBW and Siderophore-Producing Strains

The soil of the TBW rhizosphere was collected from the tobacco planting base in Datianba Town, Changning County, Baoshan City, Yunnan. The incidence rate of TBW in the field was higher than 80% (Figure 1A), and the disease index was 95%. The leaves of the tobacco plants appeared chlorotic, etiolated, and wilted at the late stage of infection (Figure 1B).

In this study, two strains of siderophore-producing fungi were isolated from the forest soil of Ailaoshan National Nature Reserve in Yunnan Province. The siderophore active metabolites of two fungi strains were effectively extracted. The experiment results show that two fungal strains, 2-8F2 (CGMCC 21047) and 2-14F2 (CGMCC 21023), showed the prominent production of siderophores (Figure 2). The activity units (SU) of strain 2-14F2 and strain 2-8F2 was 62.02% and 52.06%, respectively. The crude extracts from the solid fermentation of 2-14F2 and 2-8F2 fungi were repeatedly separated by normal silica gel and C18 reversed silica gel and purified by a gel chromatography system (Appendix A). Fungi 2-14F2 ferriferous active substances (red in CAS; red or brown in a reaction with 2% FeCl_3_) were concentrated in 3 and 5 distillates (Appendix A). 2-8f2 ferriferous active substances were concentrated in 2 and 3 distillates (Appendix A). Moreover, the 2-8F2 and 2-14F2 siderophore metabolites showed the highest absorption peaks at 420 and 423 nm, respectively (Figure 3), and the siderophore type of both strains was hydroxamate.

### 3.2. Strain Morphology and Molecular Characterization

In SEM (Figure 4), the conidia of 2-8F2 were sparsely spaced, oval in shape, 2.0–3.5 μm in length, and 1.5–2 μm in width. The surface of the conidium was folded. In contrast, the conidia of 2-14F2 were closely spaced and round with a diameter of 1.5–2.5 μm. The surface of the conidium was wavy and uneven. They were identified as *Trichoderma yunnanense* (No. 2-14F2) (Appendix A) and *Beauveria pseudobassiana* (No. 2-8F2) (Appendix A) (attachment).

### 3.3. Alpha Diversity of Soil Microbial Community

The number of bacterial and fungal OTUs that could be clustered was 5178 and 1256, respectively. The effective gene sequences of bacteria and fungi were 400–460 bp and 240–490 bp, respectively. The coverage of each treatment was more than 0.99. In alpha diversity analysis (Table 1), as compared with CK, the observed species, Shannon index, and Chao1 index of the bacterial community in the soil with 2-14F2 and 2-8F2 siderophores increased significantly (*p* < 0.05), but these indices were not significantly altered in the fungal community (*p* > 0.05).

### 3.4. Beta Diversity of Soil Microbial Community

The PCoA analysis was performed to explore the beta diversity of the bacterial community composition/structure, and the first and second principal components explain 35.93% and 17.67% of the variance, respectively (Figure 5A). In the beta diversity of the fungal community composition/structure, the first and second principal components explain 32.2% and 23.64% of the variance, respectively (Figure 5B). Compared with no application, the use of 2-14F2 and 2-8F2 siderophore-active metabolites substantially changed the fungal and bacterial community composition/structure, and different concentrations of siderophores differentially shifted the fungal and bacterial community composition/structure. The *p*-value (whether the correlation was significant) of the fungi community was higher than that of the bacteria community, indicating that the difference of the fungi community structure was more significant under the influence of different concentrations of siderophores.

### 3.5. Microbial Community Structure Impacted by Siderophore Treatment

In the bacteria and fungi community composition analysis (Figure 6), 10 dominant bacterial phyla and three dominant fungal phyla, with >1% relative abundance, were identified. The former were *Proteobacteria*, *Actinobacteriota*, *Firmicutes*, *Chloroflexi*, *Acidobacteriota*, *Patescibacteria*, *Gemmatimonadota*, *Myxococcota*, *Bacteroidota*, and *Planctomycetota*, and the latter were *Ascomycota*, *Basidiomycota*, and *Mucoromyta*. Compared with CK, the application of 2-14F2 and 2-8F2 siderophores increased the relative abundance of *Actinobacteria*, *Chloroflexi*, and *Acidobacteria*, and reduced the abundance of *Proteobacteria* and *Firmicutes*. The abundance of *Ascomycota* was increased by siderophores, and *Basidiomycota* was decreased.

LEfSe combined with LDA was used to study the effects of 2-14F2 and 2-8F2 siderophores on taxonomic groups from phyla (genus) (Figure 7). Under different treatments, the LDA score of eight bacterial phyla and two fungal phyla was greater than 2. Under 0.45 mg·mL^−1^ of siderophore metabolites, the bacterial genera with higher LDA scores were *Gemmatimonas* and *Caldalkalibacillus*. When 0.15 mg·mL^−1^ of siderophore metabolites were used, the bacterial groups with higher LDA (>2) scores were *Rhodoplanes*, while the fungal groups with higher LDA (>2) scores were *Retroconis* and *Orbiliales*. The genera *Gemmatimonas* and *Castellaniella* can be used as the indicator taxonomic groups of bacterial communities impacted by a 0.45 mg·mL^−1^ concentration of siderophore treatments, while *Chloridium* and *Acrophialophora* can be used as the indicator taxonomic groups of fungal communities impacted by a 0.05 mg·mL^−1^ concentration of siderophore treatments.

### 3.6. Enzyme Abundance of Soil Microbial Community

The PICRUSt functional prediction was conducted using the KEGG database (Figure 8). As compared with CK, the abundance of enzymes related to the metabolism redox reaction was significantly altered by the siderophore addition, and there was no significant difference between different treatments (*p* > 0.05). In the bacterial community, the abundance of aminoglycoside 3-N-acetyltransferase, glucose 1-dehydrogenase [NAD(P)^+^], 1,3-propanediol dehydrogenase, gluconate 2-dehydrogenase, GMP reductase, and chloramphenicol O-acetyltransferase increased, while the abundance of L-ascorbate oxidase, 3,4-dioxygenase, and gamma dioxygenase decreased. In fungal communities, the abundance of trypsin, aminopeptidase Y, and dipeptidyl-peptidase IV increased, and the abundance of cerebroside sulfatase decreased (*p* < 0.05). The above enzymes are related to the redox process. Our study found that, compared with no application (CK), the abundance of these REDOX-related enzymes could be significantly changed by the application of the siderophore, and the abundance of these enzymes was more significantly different than that of other enzymes. The results showed that the iron-carrier active substances in the microbial community of TBW soil could enhance the REDOX process of the microbial community.

### 3.7. BugBase Phenotypic Prediction

BugBase, as a microbiome analysis tool, can be used to determine high-level phenotypes in microbiome samples and to predict the phenotype. We found that the abundance of aerobic bacteria in the TBW soil with different concentrations of siderophore substances decreased significantly and that the anaerobic bacteria increased significantly as compared with CK (Figure 9), which may be related to the increase in redox enzyme abundance in soil by siderophore substances. The latter can significantly enhance the biofilm formation of the soil bacterial community. In TBW soil, applying different concentrations of siderophore substances significantly reduced the abundance of pathogenic, mobile-element-containing, and oxidative stress tolerant microbes. Siderophore substances could reduce the abundance of pathogenic microbes in TBW soil and regulate the microbial community structure. Under the influence of siderophore-active metabolites, the abundance of pathogenic bacteria in TBW soil decreased significantly. We thought that siderophore-active metabolites could directly inhibit the growth of plant pathogens or regulate the structure of the microbial community.

### 3.8. Effect of Siderophore on Enzyme Activity of TBW Soil

The higher the enzyme activity was, the stronger the transformation ability was. As shown in Figure 10, the activities of sucrase and urease in the soil with 2-14F2 and 2-8F2 strains were higher than those in the soil without them (*p* < 0.05). It was found that the soil sucrase and urease activities under the treatment of 2-14F2 increased with the increase in siderophore-active metabolite concentration. The soil sucrase activities under the treatment of 2-8F2 increased with the increase in siderophore concentration, while the activity of soil urease first increased and then decreased significantly with the increase in siderophore-active metabolite concentration, and the optimum siderophore-active metabolite concentration was 0.15 mg·mL^−1^. The results showed that siderophores of strains 2-14F2 and 2-8F2 significantly improved the utilization of carbon and nitrogen sources by TBW soil microbes.

### 3.9. Effects of Siderophores on Carbon Source Utilization of TBW Soil

The AWCD index reflects the number of soil microbial populations and the physiological function diversity of the microbial community to a certain extent. The higher the AWCD was, the stronger the ability of soil microorganisms to use a single carbon source was and the higher the microbial activity was. As shown in Figure 11, with the extension of culture time, the AWCD value of soil with 2-14F2 and 2-8F2 siderophore-active metabolites increased gradually, except for the 0.45 mg·mL^−1^ 2-8F2 siderophore-active metabolites, and the soil without siderophore active metabolites addition did not change significantly in the AWCD. In the period of 48 h to 144 h, the slope of the AWCD curve was the largest, indicating that the carbon source metabolic activity of soil microbes was higher at this stage. After 196 h, the AWCD curve tended to be flat, and the carbon source metabolic activity of the microbial community reached its peak. Compared with no application, the application of 2-14F2 and 2-8F2 siderophore-active metabolites significantly increased the AWCD value of diseased soil.

There were also significant differences in the types and degrees of the preferential utilization of main carbon sources by soil microorganisms under different soil treatments. As shown in Figure 12, compared with no application, the soil microbial communities with 0.05 mg·mL^−1^ and 0.15 mg·mL^−1^ of 2-8F2 siderophore, rather than 0.45 mg·mL^−1^, had a stronger utilization ability of amino acids, carbohydrates, polymers, aromatic compounds, and carboxylic acids (*p* < 0.05). By comparing the utilization ability of different carbon sources, soil microorganisms with 2-14F2 and 2-8F2 siderophores had a better utilization ability of amino acids. When compared with no application, the soil microbial communities with 0.05 mg·mL^−1^, 0.15 mg·mL^−1^ and 0.45 mg·mL^−1^ 2-14F2 siderophore-active metabolites had stronger utilization ability of all tested carbon sources. The results showed that under the influence of 2-14F2 and 2-8F2 siderophore active metabolites, the ability of the microbial community to utilize carbon sources was significantly and concentration-dependently enhanced.

## 4. Discussion

There is a large number of microorganisms that secrete siderophores in nature, e.g., *Aspergillus* [27], *Penicillium* [28], *Streptomyces* [29], *Beauveria brongniartii* [30], *Beauveria. bassiana* [31], *Trichoderma asperellum* [32], *Trichoderma harzianum* [33], *Trichoderma viruses* [34], and *Trichoderma peudokonigi* [35]. The fungi mainly synthesize and secrete hydroxamic-acid-type and carboxylic-acid-type siderophores [36]. This is mainly because most fungi produce organic acids, and the catechol-type siderophore is unstable in acidic environments, but the hydroxamate type is still very stable when the acidity is as low as pH 2 [37]. Thus, the vast majority of fungi choose hydroxamate siderophores. In addition to oxiximate siderophores, a few fungi also produce siderophores of the carboxylate type. Consistently with the results of this study, we preliminarily identified the chemical structure of the siderophores by measuring the full-wavelength absorption peaks of the metabolic products of the strains, and we found that the type of siderophore produced by the two strains was hydroxamate (Appendix A). In bacteria, these hydrophilic siderophores are composed of acylated and hydroxylated alkylamines [38]. Ambrosi found that there were a large number of carboxylate-type siderophores compounds in the metabolites of Pseudomonas fluorescens [39]. In recent years, strains secreting siderophores have become a research hotspot in the prevention and control of plant disease. On the one hand, siderophore clings to iron ions in the environment, which causes a lack of iron ions in the growth and development of pathogens; on the other hand, siderophore can be conducive to the growth of antagonistic bacteria [40]. For example, Dutta screened biocontrol bacteria and found that strains with strong control effects generally could secrete catechol-salt-type siderophores [41]. Solanki found that Rhizobium strains with siderophore activity had stronger inhibition against *Rhizoctonia solani*, while those without siderophore activity had weaker antagonistic activity [14].

TBW, as a soil-borne disease, was first caused by *Ralstonia solanacearum* invading from the tobacco rhizosphere [6]. Related studies have shown that the occurrence of soil-borne diseases mainly depends on the interactions within key microdomains [42]. Some scholars have found that the main fungi in the soil of the TBW rhizosphere were *Ascomycota*, *Zygomycota*, *Basidiomycota*, and *Chytridiomycota*. The dominant bacteria were *Proteobacteria*, *Actinobacteria*, *Acidobacteria* and *Gemmatimonadetes* [43,44]. A large number of scholars have established microbial community models based on the iron chelation principle of microbial siderophores, with the intervention of siderophores [16]. Because siderophore-producing antagonistic microorganisms have competitive advantages compared to other antagonistic microorganisms, they can regulate the community structure and composition of plant pathogens [45]. In the competition among microorganisms for iron in the soil environment, the structure of the soil microbial community can be effectively changed [26]. The results of this study indicate that strain metabolites containing siderophores promote the activity of microbial communities in TBW soil. On the one hand, the specific groups in the microbial community could have improved development under strain metabolites; on the other hand, the strain metabolites could inhibit the growth of dominant microbes and restructure the microbial community of TBW soil. In addition, during the chelation process of trivalent Fe^3+^, siderophore-active substances can effectively alter the abundance of enzymes in the redox reaction process secreted by microbial communities in the soil and increase the abundance of enzymes [46]. In this study, the metabolites of siderophore fungi were for the first time utilized to influence the microbial characteristics of TBW soil, and the results showed that the activities of sucrase and urease in the rhizosphere soil were significantly increased by siderophore-dominated strain metabolites, indicating that siderophores could improve the utilization ability of carbon and nitrogen, which agrees with the opinions of Voß et al. [47].

## Figures and Tables

**Figure 1 microorganisms-11-01535-f001:**
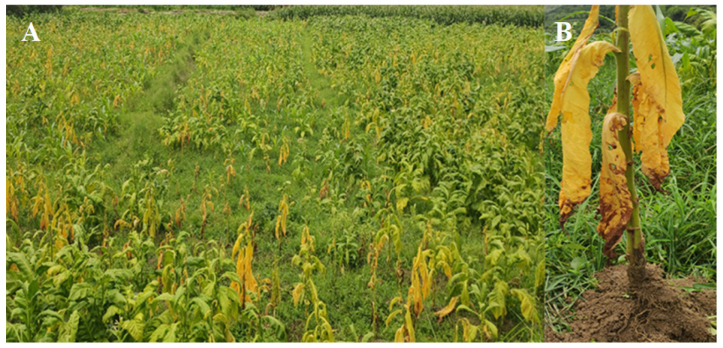
Plot of soil sampling for tobacco bacterial wilt. (**A**) TBW in the field; (**B**) rhizosphere soil of infected plants.

**Figure 2 microorganisms-11-01535-f002:**
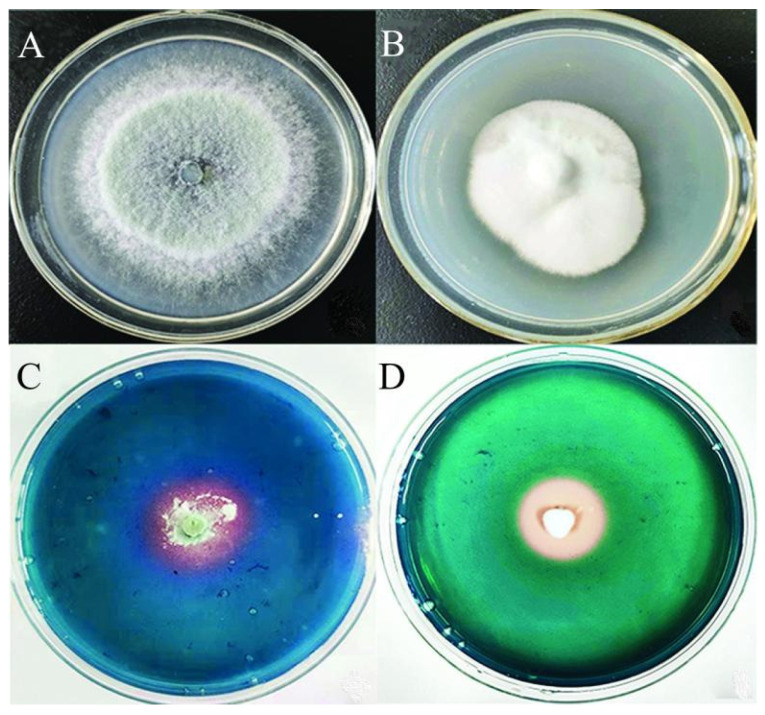
The culture morphology of 2-14F2 and 2-8F2 strains on PDA and CAS plates. (**A**,**B**) Colony of 2-14F2 and 2-8F2 strains on PDA; (**C**,**D**) 2-14F2 and2-8F2 strains were inoculated on CAS plate.

**Figure 3 microorganisms-11-01535-f003:**
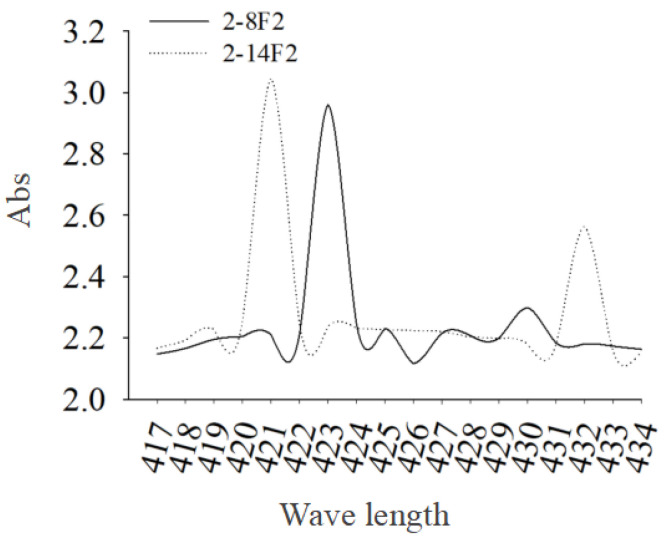
Absorbance values of 2-8F2 and 2-14F2 siderophores.

**Figure 4 microorganisms-11-01535-f004:**
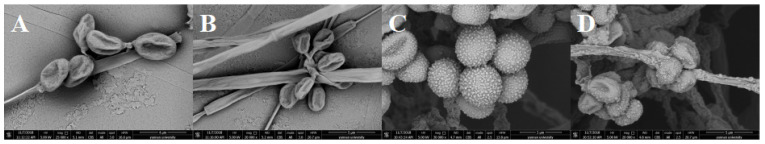
SEMobservation of fungal colony morphology and conidia morphology. (**A**) 2-8F2; ×25,000; bar: 4 μm. (**B**) 2-8F2 conidia; ×20,000; bar: 5 μm. (**C**) 2-14F2; ×30,000; bar: 3 μm. (**D**) 2-14F2 conidia; ×20,000; bar: 5 μm.

**Figure 5 microorganisms-11-01535-f005:**
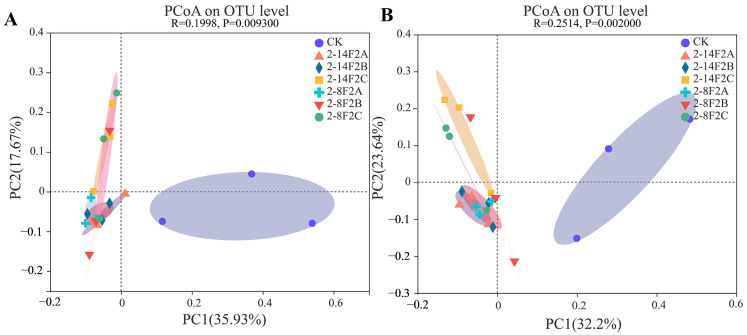
Bata diversity of the microbial community of TBW soil. (**A**) Bacteria; (**B**) fungi.

**Figure 6 microorganisms-11-01535-f006:**
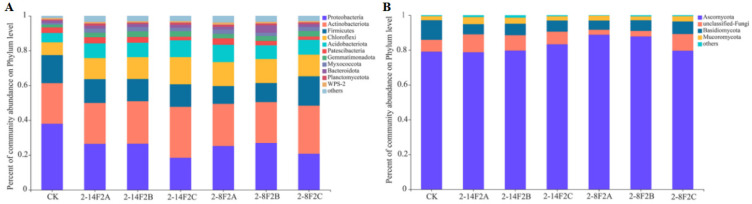
Soil microbial community structure impacted by siderophore addition. (**A**) Bacterial community composition; (**B**) fungal community composition.

**Figure 7 microorganisms-11-01535-f007:**
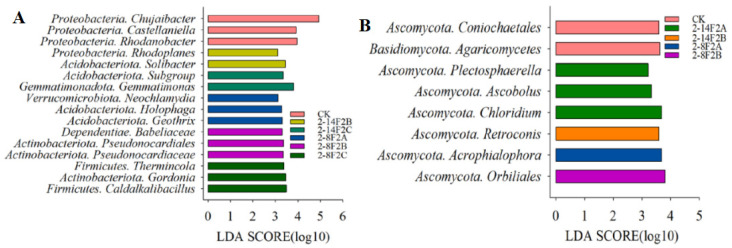
Microbial community structure of TBW soil characterized by LEfSe and LDA. (**A**) Bacterial community; (**B**) fungal community. Species under treatment LDA > 2.

**Figure 8 microorganisms-11-01535-f008:**
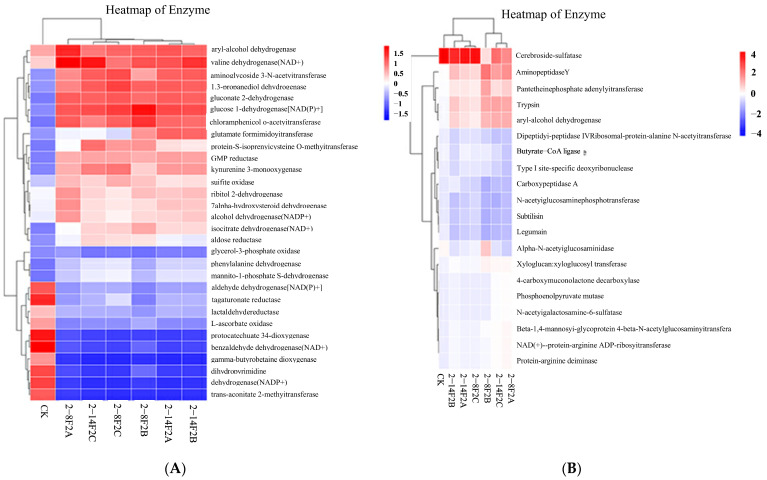
Heatmap showing the functional abundance of redox enzymes of soil microbial community. (**A**) Bacteria; (**B**) fungi.

**Figure 9 microorganisms-11-01535-f009:**
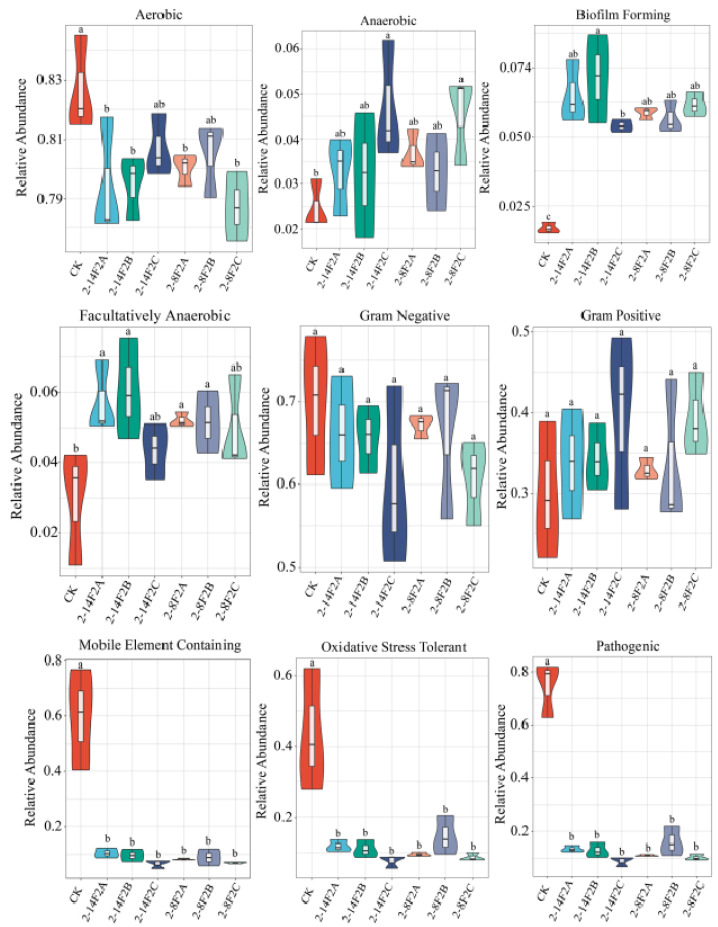
BugBase phenotypic prediction. Different lowercase letters indicate the difference of significance at the level of 0.05.

**Figure 10 microorganisms-11-01535-f010:**
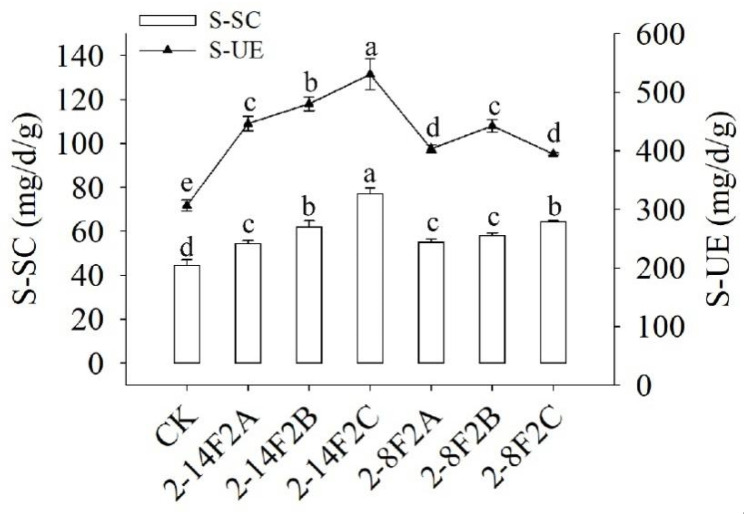
Effects of siderophores from strains 2-8F2 and 2-14F2 on soil sucrase and urease activities. S-SC is the activity of sucrase and S-UE is the activity of urease in the rhizosphere soil affected by TBW. Different lowercase letters indicate the difference of significance at the level of 0.05, which is the same as in later figures.

**Figure 11 microorganisms-11-01535-f011:**
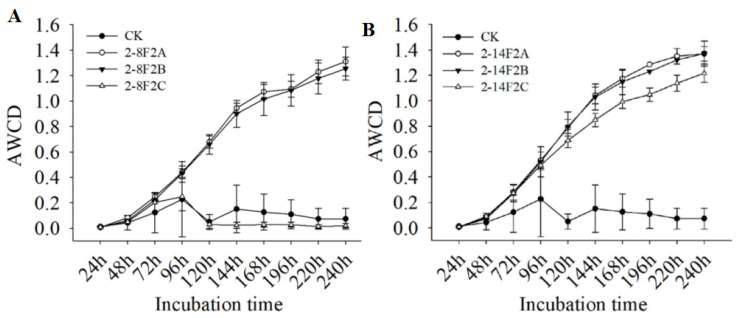
The temporal change in AWCD of the microbial community in the TBW soil sample. (**A**) The addition of 2-8F2 siderophore; (**B**) the addition of 2-14F2 siderophore.

**Figure 12 microorganisms-11-01535-f012:**
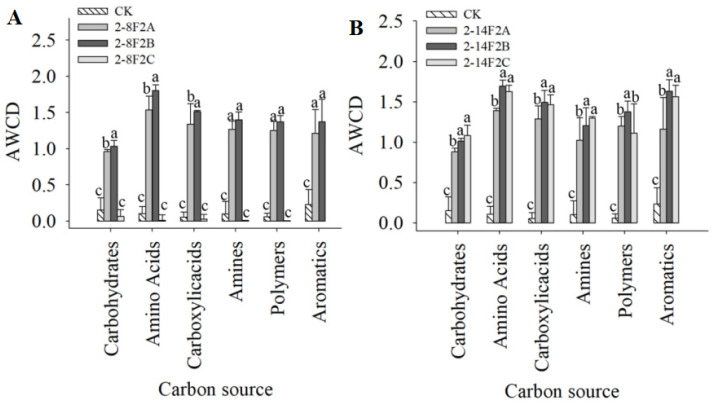
Utilization of carbon sources by soil microbial communities under different concentrations of siderophores. Different lowercase letters indicate the difference of significance at the level of 0.05. (**A**) The addition of 2-8F2 siderophore; (**B**) the addition of 2-14F2 siderophore.

**Table 1 microorganisms-11-01535-t001:** Alpha diversity of the microbial community of TBW soil.

	Index	CK	2-8F2A	2-8F2B	2-8F2C	2-14F2A	2-14F2B	2-14F2C
16S rRNA	Observed species	1586.6 (370.3) ^b^	2469.6 (86.4) ^a^	2347 (146.6) ^a^	2445 (139.9) ^a^	2435.6 (68.5) ^a^	2406.3 (131.5) ^a^	2384 (122.2) ^a^
	Shannon	4.4 (0.4) ^b^	6.4 (0.05) ^a^	6.1 (0.4) ^a^	6.4 (0.05) ^a^	6.3 (0.06) ^a^	6.3 (0.1) ^a^	6.4 (0.09) ^a^
	Chao1	2156.1 (463.7) ^b^	3334.2 (152.7) ^a^	3144.7 (149.6) ^a^	3262.7 (237.5) ^a^	3303 (130.1) ^a^	3295.5 (197.2) ^a^	3295.1 (218.9) ^a^
ITS	Observed species	329.3 (83.6) ^a^	434.6 (25.4) ^a^	422.3 (89.5) ^a^	465 (52.8) ^a^	494 (33) ^a^	479.6 (99.6) ^a^	397.6 (59.6) ^a^
	Shannon	3.4 (0.1) ^a^	3.3 (0.1) ^a^	3.2 (0.3) ^a^	3.2 (0.2) ^a^	3.5 (0.1) ^a^	3.4 (0.2) ^a^	2.9 (0.4) ^a^
	Chao1	410.8 (137.8) ^a^	565.7 (16.2) ^a^	514.9 (147.2) ^a^	574.6 (87.3) ^a^	639.8 (23.4) ^a^	608.6 (85.1) ^a^	481.3 (69.7) ^a^

The mean value (standard deviation) of triplicates is shown. Different lowercase letters indicate the difference of significance at the level of 0.05.

## Data Availability

The data sets generated for this study can be found in the NCBI under accession number PRJNA753941.

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
