# Peer review of "Differential Responses of Bacterial and Fungal Communities to Siderophore Supplementation in Soil Affected by Tobacco Bacterial Wilt (Ralstonia solanacearum)"

_microorganisms, 2023, doi:10.3390/microorganisms11061535_

Round 1
Reviewer 1 Report
The topic of the manuscript titled “Differential responses of bacterial and fungal communities to siderophores supplementation in soil affected by tobacco bacterial wilt (Ralstonia solanacearum)” is of interest for the “Microorganisms” readership, but not in the present form.
General comments:
Many sentences of the “Introduction” are written in the past, while their meaning is actual. For example, the first sentence “Rhizosphere was the key microdomain of…“ let the reader understands that now the rhizosphere is something different from the microdomain of plant-soil-microbe interactions.
The “Discussion” is poor and full of sentences that, substantially, repeat concepts already reported in other parts of the paper. For example, at lines 485-488, the sentence “In this study, two fungal strains, Trichoderma yunnanense 2-14F2 and Beauveria pseudobassiana 2-8F2, with high siderophore activity were obtained by screening a large number of strains from forest soil of Ailao Mountain in Yunnan, China.” can be deleted. The same for the sentences from line 518 (In this study…) to line 523 (…and other methods.). Rather, Authors should discuss their results.
Many references (1, 10, 11, 16, 32, 33, 34, 43 for sure) state the Authors' full name instead of the last name. Please check carefully.
Specific comments
Lines: comment
17-20: Please check the sentence “The influence of…bacterial wilt (TBW).” It seems that it lacks the verb.
21: It should be “average well color development” and not “average pore color development”. Please check and correct, if necessary.
47: Please, write the extended words for the acronym SPMs.
50: Please check the sentence: “…, iron was rich…” This reviewer suggests changing as follows: “soil and crust are rich in iron, an essential element for plants and microorganisms”.
114-116: The range of soil organic matter is very large. Why? In addition, Authors should mention the methods used for the determination of the organic matter, moisture, iron mass and all the other physical and chemical soil parameters presented in this section of the paper.
125: “soil sample” is written twice. Please check. In addition, ml must be reported as mL. Please correct here and throughout the paper.
126: coical bottle? Please check.
144: add a space between culture and were
179: Please check “ferriderite”. Did you mean ferrihydrite?
202: add a space between mL and 30
281: It should be as follows: “Average well color development (AWCD) index…
298: The sentence should be as follows: “The leaves of tobacco plants appeared chlorotic, etiolated and wilted…”
323: Delete “he” before “conidia”.
337: The sentence should be as follows: “…siderophores increased significantly…”
431: “…0.05). It was…
445: delete “(average…development)” and add “index” after AWCD.
535-536: please check “oxiximate”. Is it right? Did Authors mean hydroxamate?
Please improve the English language. I would encourage you to have the technical revision read by a native English speaker before you resubmit.
Reviewer 2 Report
Dear Authors
The research topic is quite interesting, and the outcomes are commendable.
Comments for authors
Lines 23 and 25 should be rewritten precisely.
Line 25: enhance LEfSe
Line 63: include relevant references
Lines 67 and 68: rewrite exactly
Rewrite lines 88-90 so that the sentence highlights the purpose of the study.
Material and methods Rewrite section 2.6 in its entirety.
Rewrite the title of Figure 1.
Figure 4 illustrates the required distance between SEM, observation, and scale.
Line 337 requires a space before'significantlincreased' 490-492
confirm the citation in section 526 of the results.
In addition, the materials and methods section was quite complicated and unclear.
Authors should update their references following the MDPI journal format.
I recommend language editing and double-checking the space between text and citation throughout the manuscript.
Round 2
Reviewer 1 Report
The topic of the manuscript titled “Differential responses of bacterial and fungal communities to siderophores supplementation in soil affected by tobacco bacterial wilt (Ralstonia solanacearum)” is of interest for the “Microorganisms” readership.
General comments:
The manuscript has been improved with respect to the previous version.
Only few comments
Lines: comment
16: I suggest changing “The siderophore” in “Siderophores”.
20: It should be “was” instead of “were”. Please check.
64: “was” should be “is”. Please check.
490: Change “oxiximate” with “hydroxamate” or, if it is right, provide at least a reference.
Reviewer 2 Report
Dear Authors
There have been modifications and corrections made to the language. However, the researcher and scientific community will find this manuscript useful; best wishes for your future research.
